# Can Different Parameter Sets Lead to Equivalent Optima between Geometric Accuracy and Mechanical Properties in Arburg Plastic Freeforming? [note 1]

**DOI:** 10.3390/polym15061516

**Published:** 2023-03-18

**Authors:** Lars Eisele, Anselm Heuer, Kay A. Weidenmann, Wilfried V. Liebig

**Affiliations:** 1Institute for Applied Materials—Materials Science and Engineering (IAM-WK), Karlsruhe Institute of Technology (KIT), Kaiserstrasse 12, 76131 Karlsruhe, Germany; utwgb@student.kit.edu (L.E.); wilfried.liebig@kit.edu (W.V.L.); 2Institute of Materials Resource Management (MRM), University of Augsburg, Am Technologiezentrum 8, 86159 Augsburg, Germany; kay.weidenmann@mrm.uni-augsburg.de

**Keywords:** density measurement, roughness, porosity, µCT, analytical model

## Abstract

Technological advances have led to the increased use of plastic-based additive manufacturing processes for the production of consumer goods and spare parts. For this reason, the need for the best possible mechanical properties while maintaining geometric accuracy is becoming increasingly important. One of these additive manufacturing processes is the Arburg Plastic Freeforming process, which differs from the widely used Fused Filament Fabrication process in the way that droplets are discharged along a track instead of continuous extruded tracks. As with all other plastic-based additive manufacturing processes, due to the round shape of the tracks, voids occur between the individual tracks during manufacturing, which effects mechanical properties. In contrast to previous work, which mainly focused on how the mechanical properties change with a change in a single printing parameter, this work focused more closely on the interaction of three relevant printing parameters considered as a parameter set. Their influence on the mechanical properties was investigated by tensile tests, the influence on the residual porosity by density measurements and the influence on the geometric accuracy by surface roughness measurements. It was shown that by considering the parameters as a parameter set, states of high density and therefore high mechanical properties while reaching minimal surface roughness can be achieved for significantly more combinations than previously assumed. However, for these states the residual porosity was slightly different. This difference was explained by a parameter-dependent deformation factor of the droplets, which influences the maximal possible degree of filling during manufacturing. For the optimization of arbitrary parameter sets, an analytical model was derived.

## 1. Introduction and Motivation

Increasingly precise technologies, the possibility of using an expanding range of materials and many more factors have contributed to the fact that plastic-based additive manufacturing is no longer only used in model and prototype design, but is also being applied in the production of consumer goods and functional parts in numerous industries. Usually, compared to other manufacturing processes such as injection moulding, plastic-based additive manufacturing processes have lower mechanical properties [1,2,3,4]. This phenomenon is caused by the void formation that occurs at the joints of the individual tracks due to their round shape. According to Tronvoll et al., these voids cause incomplete polymer chain diffusion between the individual tracks, decrease the cross-section, and increase the notch effect, so that part strength and stiffness are negatively affected depending on the size and shape of the voids [5]. An attempt is therefore made to minimize these void effects by means of suitable parameter settings and thus to achieve mechanical properties equivalent to those of the bulk material. For this reason, the influences of certain parameter settings on the mechanical properties of parts have already been intensively investigated for plastic-based additive manufacturing processes.

One of these processes is the Arburg Plastic Freeforming (APF) process. Characteristically, plastic granules are plasticized in an injection moulding machine followed by a high-frequency closable nozzle forming droplets. These droplets join together and form a droplet chain [6,7]. It was found by Charlon et al. that, out of numerous parameters that can be adjusted in this process, the most important for the mechanical properties are those that influence the heat input into the material and the orientation of the tracks within the layers [8]. With regard to the orientation of the tracks, a ±45° configuration, which means a positioning of the droplet chains at an angle of 45° to the direction of the force, rotated by 90° from layer to layer, proves to be the most suitable for high mechanical properties [3,8,9]. This is due to the fact that droplet chains on adjacent tracks can be placed next to each other within a very short time, so that they have not yet cooled down much and consequently better join together [8]. It has also been shown that the mechanical properties of the manufactured part improve when the nozzle temperature is increased. This temperature increase reduces the viscosity of the material and lowers the void volume due to better distribution of the plastic. However, to avoid degradation of the plastic, which would be accompanied by a reduction in the mechanical properties, the nozzle temperature cannot be set arbitrarily high.

Besides the heat input and the raster orientation, the parameter influence on volume-filling of the part must be taken into account. For the APF process the volume-filling within one layer depends on the layer thickness (LT), the distance between the tracks within the layer and the droplet volume itself, which is controlled by the discharge rate (DR). The distance between tracks is normally equated with the track width and can subsequently be determined by the droplet aspect ratio (DAR), which is defined as the ratio of track width to layer thickness. If all other parameter settings remain unchanged, an increase in LT leads to a larger void volume and lower tensile properties [3,8]. Charlon et al. explain this by the increased distance of the nozzle to the already manufactured part, which reduces the heat input of the nozzle into the material and thus the cohesion between the layers [8]. Although they pointed out that the volume that needs to be filled within one layer increases by increasing LT, no other parameters were adjusted. In relation to the DAR, it was shown that the mechanical properties are higher when the DAR is reduced [3,6,8,10]. Hirsch et al. justify this by a higher degree of filling because of the smaller distance between the tracks within the layers [3]. According to the state of the art, the lower the LT and the lower the DAR, the better the mechanical properties. Nevertheless, the volume-filling within a layer depends not only on these two parameters, but also on the DR. Charlon et al. point out that an increase in DR leads to higher part density and tensile properties [8]. In summary, some of the mentioned researchers point out that high mechanical properties are associated with high part density.

Previous work has demonstrated that reducing LT or DAR from a starting point increases part density and thereby improves mechanical properties. This is mostly explained by the reduction in the volume that needs to be filled within a layer or by the increase in the number of tracks within a layer, respectively. Since DR was not changed in these investigations, more plastic was effectively used to manufacture parts. Furthermore, previous work has mostly focused on how the mechanical properties change when a single parameter is varied. In this work, a different approach is used to investigate the influence of LT, DAR, and DR on part density and mechanical properties. First, LT and DAR are considered as geometric influencing parameters that define the volume-filling within a layer. Then DR is varied for different LT-DAR combinations to change the volume flow of the machine in order to fill the desired volume within a layer. In summary, LT, DAR, and DR are considered as a parameter set. The aim of this approach is to investigate whether an increase in LT or DAR or a reduction in DR can also increase the part density and thus improve the mechanical properties when the other parameters of the parameter set are adjusted. In this case, the previous findings on the parameter influences are confronted with further operating principles. As a second aim of this work, geometric accuracy is taken into account as an additional boundary condition. Charlon et al. noted that by increasing the DR too much, too much material is pressed into the part, which consequently flows outward resulting in an inaccurate shape of the part [8]. This is considered as overfilling and is not desirable in technical applications. The relevance for maintaining geometric accuracy is shown, for example, in the investigation of Hecker et al., which focused on shrinkage behaviour in the APF [11]. In the present work, a process setting with maximum part density while maintaining geometric accuracy is considered as an optimal degree of filling. It is assumed that maintenance of geometric accuracy can be assessed by the surface roughness. With the measurement of the surface roughness, the optimal degree of filling for different parameter sets is determined, which enables the investigation of whether there is always an equivalent density at the optimal degree of filling or whether the density differs depending on the parameter set. X-ray computed microtomography (µCT) is used to further investigate the residual porosity at the optimal degree of filling. For these specific parameter sets, the tensile properties are also determined and compared. To the best of our knowledge, the present work provides the first comprehensive investigation of the optimal degree of filling in the APF process. The results highlight an interesting correlation between the deformation of the droplets in the manufacturing process and the residual porosity. Finally, for the optimization of arbitrary parameter sets, an analytical model is derived which can be also used to determine an optimal degree of filling.

## 2. Material and Methods

### 2.1. Arburg Freeformer and Slicing Software

The specimens were additively manufactured by using the APF process on the freeformer 200-3X from ARBURG GmbH+CoKG in Germany. As shown in Figure 1a, the machine has a plasticating cylinder with an inner diameter of 15 mm, which can be heated by two heating zones. First, the plastic granule is plasticized and then dosed against a stagnation pressure by using a screw. Before the plastic is discharged from an also heatable nozzle, a decompression is carried out for closing of the non-return valve. After the melt cushion has been reached, dosing is repeated. The diameter of the nozzle is 0.2 mm. This nozzle can be opened and closed at variable frequencies so that the plastic is discharged in individual droplets that combine to form droplet chains. In total, the freeformer has two of these discharge units, consisting of a plasticating cylinder and a nozzle. The droplet chains are deposited in a heatable chamber containing a build plate that can be moved along three axes. The machine is an open system which allows the part properties to be influenced by parameter adjustment. Before manufacturing a part with the freeformer, a model of the part must be processed in the slicing software. ARBURG freeformer software v2.30 was used in this work. This software slices the model into individual horizontal layers depending on the parameter LT set by the user and creates tracks within the layers along which the droplet chains are placed. Figure 1b provides a visualisation of the relevant parameters for filling a layer. If the filling density (referred to as infill degree in the FFF-process) for a part is set to 100%, the slicing software generates tracks at a distance *T* equal to the track width *W*. The APF process specifies a new parameter DAR, defined as
(1)$$\begin{array}{c} DAR=\frac{W}{LT}, \end{array}$$
which must be set by the user in the slicing software. Accordingly, the distance between the tracks *T* is set to
(2)$$\begin{array}{c} T=W=DAR\cdot LT. \end{array}$$

Some researchers explain the choice of an appropriate DAR by an experimental characterisation of the droplet chain geometry [12,13]. In this case, the droplet height is considered as LT and the droplet width as track width *W*. This method is not used in this work. The third parameter DR can be used to adjust the volume flow of the freeformer, since the droplet volume shown in Figure 1b is proportional to the DR [8]. DR is defined as a parameter in percent and can be described by the travel increment *L* of the screw related to a reference travel increment $L_{0}$. To ensure that leakage at the non-return valve does not affect this relationship, DR is controlled internally using other control parameters [14]. It should be noted here that the variation of DR also influences the droplet geometry, and if the previously mentioned method is used to determine the DAR, DR consequently influences the DAR as well. This is a possible reason why in some previous works DR was not further investigated and kept constant. In summary, the three parameters LT, DAR, and DR can be used to influence the volume-filling within a layer.

### 2.2. Material Used

All specimens were manufactured with acrylonitrile–butadiene–styrene (ABS) Terluran GP35 granules from INEOS Styrolution. According to the data sheet, the bulk density of the material is 1.04 g/cm^3^. This value was confirmed experimentally. The tensile modulus is 2300 MPa, the tensile strength is 44 MPa and the heat deflection temperature at 0.45 MPa is 95 °C [15]. The granules were dried at 80 °C for 2.5 h.

### 2.3. Manufacturing of the Specimens

To investigate the influence of the parameters on the degree of filling, specimens for density measurement and surface analysis were manufactured. These specimens were cuboids of base area 16 × 20 mm and a height of 25 mm. Five different combinations of LT and DAR were selected according to Table 1. For any of these five combinations, the specimens were fabricated at five different DRs each. For example, for a LT of 150 μm, a DAR of 1.4, and DR of 25%, the nomenclature LT150DAR14DR25 is used. The DR was selected in such a way that for each combination cuboids were available in a degree of filling range from underfilled to overfilled. For statistical purposes, five identical cuboids were manufactured for each DR value. In total, 25 cuboids were produced for each LT-DAR combination.

Other cuboids were made to measure the surface roughness in the region of the optimal degree of filling (optDF) previously introduced. The geometry of the cuboids was not changed. For each of the five LT-DAR combinations introduced in Table 1, five new cuboids were manufactured whose DR differed by one percent each. The range of the DR can be found in Table 2. Within this range of DR, a cuboid was selected for measurement of the density using the principle of Archimedes and investigation of the porosity using µCT. The selected DR can also be found in Table 2. As the specimens from Table 1, all other specimens from Table 2 where manufactured one by one.

For characterisation of mechanical properties, ten dog-bone specimens were manufactured for each LT-DAR combination with the selected DR from Table 2 according to DIN EN ISO 527 Type 1B. Due to the high number of specimens and since Hirsch et al. have shown that the manufacturing of multiple specimens per building job has no influence on the geometry of the voids, the entire build plate was used [16]. Accordingly, seven dog-bone specimens could be manufactured at the same time.

Slicing process was the same for all specimens. The perimeter was set to one, the raster orientation was set to 45°, overlap was set to 50%, and the printing speed was set to 20 mm/s in the perimeter and 65 mm/s in the filling. The build plate moved at 250 mm/s. For the filling density 100% was chosen. The swelling factor was set to one in the slicing process. The machine parameters for manufacturing the specimens are given in Table 3.

### 2.4. Density Measurement

The principle of Archimedes was used to determine the density of the specimens. In order to prevent the water in which the density is measured from entering the microstructure and falsifying the measurement, preliminary tests were carried out. Whenever the density of the specimens was greater than 0.95 g/cm^3^, the surface structure was close enough to prevent water from entering the microstructure. If the density was below 0.95 g/cm^3^, specimens were embedded in VariKem 200 from Schmitz Metallographie GmbH. It was placed in small silicone moulds in which the specimens were immersed after 180 s. After this time, the viscosity of the embedding medium was high enough to prevent it from entering the microstructure and low enough to net the surface.

Figure 2 shows a microscopic cross-section of an embedded and underfilled specimen. The embedding medium can be seen at the bottom and in the right area of the cross-section. There is no embedding medium in the microstructure. The density of these embedded specimens ρ_Specimen_ can be determined by
(3)$$\begin{array}{c} \rho_{\mathrm{Specimen}}=\frac{m_{\mathrm{Specimen}}}{V_{\mathrm{Total}}-V_{\mathrm{VariKem}}}. \end{array}$$

The mass of the specimen *m*_Specimen_ can be determined using a weighing scale before embedding. The total volume *V*_Total_ can be determined after embedding using the principle of Archimedes. From the difference in mass of the embedded and not embedded specimen, the volume of the embedding medium *V*_VariKem_ can be calculated with the previously determined density of the embedding medium. This density was determined on three specimens after curing from the embedding medium in the silicone mould. The mean density is 1.204 g/cm^3^ and could be precisely measured with a standard deviation of 0.00058 g/cm^3^.

To measure the density of the dog bone specimens, a piece of 2 cm length was cut out from the narrow region of each specimen. These pieces were not embedded before density measurement.

### 2.5. Surface Structure by Optical Coherence Tomography (OCT)

A 3D reconstruction of the top surface structure was obtained by using the TELESTO Series Spectral Domain OCT Imaging System TEL320C1 with the Scan Lens LSM03 from Thorlabs GmbH and evaluated in the software ThorImageOCT 5.4.2. The field of view was set to 10 × 10 mm with a pixel size of 12 μm. Speed of the scanning system was set to 10 kHz. The 3D reconstructions were acquired with the OCT in 3D mode. With the help of these 3D reconstructions, underfilling and overfilling can be investigated in more detail.

### 2.6. Surface Roughness Measurement

As a parameter for the surface roughness, the arithmetical mean height Sa was determined by measuring the surface structure with a µsurf confocal microscope from NanoFocus AG and evaluating it in the µsoft analysis software v7.2 developed by the manufacturer. According to DIN EN ISO 25178-2, Sa is defined as the mean difference in height from the mean plane. A section of 8.35 × 8.06 mm at the top surface of a specimen was scanned starting from the same reference position. The section was at least 2.5 mm away from the edges of the surface. The acquired surface structure was first processed in the software in such a way that an unwanted inclination due to unevenness was calculated out by means of least-squares. Subsequently Sa was evaluated for each specimen. For each LT-DAR combination, the DR with the lowest Sa was used as the basis for the dog-bone specimens to be printed.

### 2.7. Porosity Characterisation with X-ray Computed Microtomography (μCT)

Slice images and 3D reconstructions of specimens were received by using a YXLON Precision µCT-scanner. The acceleration voltage of 100 kV and target current of 0.12 mA were used. A total of 2400 projections were recorded on a Perkin Elmer XRD1620 AN flat panel detector with 2048 × 2048 pixel. For noise reduction exposure time was set to 800 ms. The obtained projections were used for reconstruction, which was carried out in VGStudio MAX 3.4 using the FDK algorithm. The voxel resolution of the 3D reconstructions was 20.7 μm. In VGStudio MAX, a surface detection and subsequent porosity measurement was performed with VGEasyPore. A relative algorithm was used to determine the threshold between material and air.

### 2.8. Tensile Tests

Mechanical properties of specimens were determined from tensile tests on a universal testing device Zwick/Roell ZMART.PRO 200 kN with a 20 kN load cell. Hydraulic jaws with a contact pressure of 10 bar were used to grip the specimens on both sides. Specimens were tested at a crosshead speed of 2 mm min^−1^ and an initial tension of 0.5 MPa. The tensile modulus was measured directly on the specimen in the range of 0.25% to 0.5% strain by using a extensometer Zwick/Roell multiXtens BTC-EXMULTI. All specimens were tested in a conditioned laboratory at 23 °C.

## 3. Results

### 3.1. Density of Specimens for Different Parameter Sets

The density above the DR are shown in Figure 3 for each LT-DAR combination from Table 1. Figure 3a shows the density for variable DAR at constant LT, while Figure 3b shows the density for variable LT and fixed DAR. LT200DAR14 can therefore be found in both figures. The mean standard deviation for all parameter sets is 0.009 g/cm^3^, the minimum standard deviation is 0.001 g/cm^3^ (LT150DAR14DR40), and the maximum standard deviation is 0.032 g/cm^3^ (LT200DAR11DR30). It is observed that the standard deviation of the density tends to decrease as the DR increases. It is noticeable that the density for each LT-DAR combination increases almost linearly with increasing DR, until it finally flattens out for the highest DR used for each combination. It can be observed that for increasing LT and increasing DAR, the DR must also be increased to manufacture specimens with the same density.

It is also noticeable that despite the same DR range of 20% for each LT-DR combination, the range of density for these combination produced at higher DR decreases continuously. For LT200DAR11 with a DR range of 30–50%, the highest and the lowest density are 0.31 g/cm^3^ apart, while for LT200DAR17 with a DR range of 95–115% this difference is almost 50% lower at 0.16 g/cm^3^. Related to this observation is the finding that the slope in the linear region of the curve decreases with increasing DR. The slope for LT150DAR14 is 0.017 (g/cm^3^)/% while for LT250DAR14 it is 0.007 (g/cm^3^)/%.

### 3.2. 3D Reconstruction of Top Surface for LT200DAR14

Surface structures from the top of the specimen for the LT-DAR combination LT200DAR14 are shown in Figure 4. For each DR used for this combination, there is a 3D reconstruction of the surface structure. These 3D reconstructions were determined with OCT. For the lowest DR of 60%, the individual droplets and droplet chains can be identified along the track. Between these tracks, gaps can be seen where no material is present. If the freeformer had deposited another layer on this top surface, voids would appear in those regions where gaps occur. The surface structure of the specimen manufactured at 65% DR is comparable to the one made at 60%, but the gaps between the tracks are smaller. This trend continues as the DR increases, so that only very small gaps between the tracks can be seen in the surface structure of the specimen produced with 70% DR. At a DR of 75%, the surface structure appears dense and the individual tracks touch each other, so that no more gaps are visible. Above a DR of 80%, the tracks can no longer be clearly identified as they merge into each other. The surface structure appears less rough and homogeneous. However, it is noticeable that it is no longer flat but curved. Thus, a global waviness can be detected.

### 3.3. Arithmetical Mean Height Sa for LT-DAR Combinations

Figure 5a–e show the arithmetical mean height Sa above a range of 5% DR for the five LT-DAR combinations from Table 2. The range of DR was chosen based on surface structure investigations with OCT. It is recognizable that a minimum of Sa exists for all combinations, while Sa for LT150DAR14, LT200DAR11, and LT200DAR17 initially increases a little with a DR increase before it finally reaches the minimum. Sa decreases continuously for LT200DAR14 and LT250DAR14 until it reaches the minimum. Subsequently, Sa increases again for all LT-DAR combinations. Overall, Sa ranges from about 5 μm to slightly below 25 μm, depending on the LT-DAR combination and the DR. Figure 5f shows Sa for all combinations in one graph. It can be seen that the minimum of Sa for a combination increases with increasing DR and that for a constant LT the minimum of Sa is almost identical.

### 3.4. Density and Porosity at Optimal Degree of Filling

For each LT-DAR combination from Table 2, a specific DR was chosen for a more detailed investigation of the residual porosity. The basis for the choice was the minimum of the arithmetical mean height Sa from Figure 5. At this minimum, optDF was assumed to be present for the parameter sets. For these parameter sets, Table 4 shows the density and porosity measured with the principle of Archimedes and the porosity measured with VGEasyPore based on a µCT-Scan.

Except for the parameter set LT200DAR11DR45, all densities are above 1 g/cm^3^. There is no observable trend between the density and the DR, and also not between the density and LT. However, as the DAR increases, the density increases. The porosity of a parameter set differs slightly depending on the measurement method. This difference is in the range of 0.25–1.77% (absolute percentage points). However, if the parameter sets are sorted by the measured porosity, their order is identical for the measurement according to the principle of Archimedes and the measurement with µCT.

Slice images of specimens with the specific parameter sets from Table 2 are shown in Figure 6. These slice images were acquired by analysis of µCT reconstructions. Due to the fact that the specimens have warped slightly during manufacturing, the flat slicing plane cannot cut exactly between two layers. Therefore, some regions appear less porous in this slicing plane. Within a layer, the porosity for all parameter sets is negligible and voids are only present between the filling and perimeter. Between two layers, the amount of voids qualitatively corresponds to the height of porosity from Table 4. In this slicing plane, the voids are more or less evenly distributed at the interfaces between neighbouring tracks. The strikingly small porosity of LT200DAR17DR113 can also be seen from the slicing planes. In the upper slicing plane, voids between the layers are no longer visible in some cases. In comparison, the voids in the LT200DAR11DR45 are visible at almost every interface between the tracks.

### 3.5. Mechanical Properties at Optimal Degree of Filling

Figure 7a,b shows the tensile strength and tensile modulus for the parameter sets from Table 2, which were selected for the investigation of optDF. These parameter sets correspond to those investigated in the last chapter. It can be seen, that tensile strength and tensile modulus increase with increasing DAR at constant LT. At constant DAR and different LT, the tensile strength and tensile modulus remain nearly at the same level of *σ_m_* = 30 MPa and *E* = 1150 MPa. LT200DAR17DR113 has the highest mean tensile strength of *σ_m_* = 32.9 MPa and the highest mean tensile modulus of *E* = 2261.9 MPa. Based on the tensile modulus from the ABS data sheet, injection moulding results are almost achieved. In comparison, LT200DAR11DR45 exhibits the lowest mean tensile strength of $\sigma_{m}=27.9$ MPa and a mean tensile modulus of E=2030.2 MPa. It is still noticeable that the standard deviation for the tensile strength is nearly similar for all parameter sets with 1.1 MPa, while it is nearly twice as large with 1.99 MPa for LT150DAR14DR32. This is also valid for the tensile modulus with the difference that the standard deviation decreases further with increasing DR. Representative stress–strain curves for the five parameter sets are shown in Figure 7c. The differences in tensile strength and tensile modulus between the parameter sets are clearly visible. The elongation at break is subject to greater scattering due to small differences in the clamping procedure, so that it is not suitable for characterising the mechanical behaviour.

Figure 7d shows the density of the dog-bone specimens for the parameter sets from Table 2. A correlation between density and mechanical properties can be observed. The density also increases with increasing DAR at constant LT. In comparison, a slightly trend can be observed for an increasing LT at constant DAR. The highest mean density for the dog bone specimens are also obtained from parameter set LT200DAR17DR113 and the lowest for LT200DAR11DR45. The largest standard deviation also occurs for the parameter set LT150DAR14DR32, but the difference with the other parameter sets is smaller.

## 4. Discussion

### 4.1. Influence of the Parameters LT, DAR and DR on the Degree of Filling

The fact that the density increase as a function of DR in Figure 3 barely differs from LT-DAR combination to LT-DAR combination shows that by varying the parameters LT, DAR, and DR, parts with different porosity and density, respectively, can be manufactured in a targeted manner. A part with a certain density can be manufactured for any combination of LT and DAR as long as the DR is accordingly adjusted. This observation has implications for the previous understanding of parameters in the APF. For example, according to Hentschel et al., a decrease in DAR leads to an increase in density and thus to an increase in mechanical properties [10]. Or according to Charlon et al., a decrease in LT also leads to an increase in density and thus to an increase in mechanical properties [8]. It was concluded that the DAR and LT must be reduced for high density and mechanical properties, but this is not necessarily the case when DR is taken into account. The three parameters LT, DAR, and DR are rather to be seen as a parameter set for controlling the degree of filling or density within a layer and must be adjusted to each other for a desired density. In this case, LT and DAR could remain constant at any level and only by adjusting DR, the same density can be reached compared to a reducing of LT or DAR. In summary, the observation from Figure 3 contradicts the statement that high density and mechanical properties can only be achieved with low LT and low DAR.

Besides the free adjustability of the density, a clear linear region between the density and the DR below 1.02 g/cm^3^ is noticeable in Figure 3. This behaviour can be explained as follows: The outer contour of the specimens is defined via the perimeter and due to the surface sealing (see Figure 2) no water can enter the microstructure during the density measurement. As a result, the total volume of the specimens remains constant. For the volume flow of the freeformer, ${\dot{V}}_{\mathrm{freeformer}}∼$
*DR* is assumed and under the assumption of a constant density of the discharged droplets, ${\dot{m}}_{\mathrm{freeformer}}∼\;DR$ applies. Therefore, with a linear increase in mass, the density of the specimens also increases linearly as long as the specimen is underfilled and additional mass can be added to the microstructure. This relationship remains as long as the total volume of the specimens is not significantly changed by the increase in DR.

It is noticeable that the bulk density of ABS (1.04 g/cm^3^) is barely or not at all reachable. Not even when the DR is increased to any value within a LT-DAR combination, which is due to the fact that the density curves flatten out and the linear relationship between the density and the DR is lost. Specimens close to the bulk density of ABS also show significant overfilling as shown in Figure 8 for the LT-DAR combination LT150DAR14 and no longer meets the requirements for additively manufactured parts.

This implies that, as in the FFF process, a part manufactured by the APF process always has a certain void volume in the microstructure, which is due to the way a part is additively manufactured. The individual rounded tracks must first fuse together and the necks grow between the tracks. However, the material solidifies before complete coalescence, resulting in partial voids between the tracks [17,18]. A further increase in the DR cannot fill these voids and will inevitably lead to overfilling. This state of overfilling is not desired and not acceptable from a procedural point of view.

### 4.2. The Determination of an Optimal Degree of Filling

In the present work, a state is defined as optDF if the void volume is small, the density high, and the external geometry is still accurately reproduced. This state should exist for each LT-DAR combination and should be located at the transition between underfilling and overfilling. On the basis of the 3D reconstructions of the surface structure in Figure 4, underfilling and overfilling can be qualitatively comprehended. The large gaps, which could be seen between the tracks at a DR below 70%, reduce the density of the specimen. The specimen is underfilled because its microstructure would allow more material to enter. To fill up these gaps and voids in the microstructure and consequently increase the density, more material has to be introduced into the specimen by increasing DR. Theoretically, due to the gaps and the resulting depressions, the surface roughness should be higher at an underfilling than for an optDF. At a certain point, however, the void volume becomes so small that material does not flow exclusively into the voids, but also to the outside. Overfilling begins with the onset of waviness, as seen in Figure 4e for LT200DAR14DR80 or in Figure 8 for LT150DAR14DR35. This overfilling breaks the boundary condition of geometric accuracy, which is why optDF must be located at smaller DR. An overfilling does not lead to an even accumulation of material, as the nozzle, for example, shifts material excess in edges and leaves it at turning points. A theoretical increase in surface roughness can be derived from this behaviour. To summarise, surface roughness should be minimal at optDF compared to underfilling and overfilling.

This theoretical consideration is in good agreement with the behaviour of the arithmetical mean height Sa over the DR for all LT-DAR combinations in Figure 5. Each combination has a minimum of Sa and for LT200DAR11, LT200DAR14 and LT250DAR14 this minimum can be determined clearly. For LT150DAR14 and LT200DAR17, the behaviour in case of underfilling is unusual at first, but can be explained by the measurement method. Due to gaps between the tracks, in some cases the confocal microscope does not find a depth value or the value is not correct, leading to statistical variations in case of underfilling. Previously, a *good* degree of filling was qualitatively identified on such surface structures as shown in Figure 4, as Hentschel et al. conducted for example [19]. The method used in this work can be considered more reliable.

Comparing the DR values at minimum Sa (32%, 45%, 75%, 113%, and 147%) from Figure 5 for each LT-DAR combination with the corresponding density curves from Figure 3, it is noticeable that the DR values are still in the linear region. This leads to the conclusion that the onset of overfilling is still in the linear region and that the density behaviour above the DR is therefore not suitable for finding optDF. The reason for this could be that although the total volume of the specimen is increased at the onset of overfilling, there is still some material being pressed into the voids.

In summary, for each LT-DAR combination, it is possible to determine a unique parameter set for manufacturing a part with optDF. For the investigated combinations used in this work, the DR for optDF can be taken from the green marked points in Figure 5.

### 4.3. Model for Calculating a Desired Degree of Filling

An optDF is of great importance for the application of additive manufacturing processes such as the APF process. Until now, however, suitable parameters for the APF process could only be determined retrospectively, which made the finding of suitable parameters a laborious process. Especially, when deviating from proven standard parameters. For this reason, a model is proposed to calculate the degree of filling for combinations of LT, DAR, and DR in the linear region of density increase.

As described in Section 2.1, the freeformer can only manufacture parts based on the file created by the slicing software, which contains information about the parameters and tracks used. For the APF process, there are two volume flows with regard to the tracks: one that results from the slicing software and one that is adjusted on the hardware side (freeformer) during the actual manufacturing process. These two volume flows are derived and equated in the following for inflexible volumes in order to be able to calculate the required parameters for a desired degree of filling.

The ideal volume of a droplet from the slicing software is illustrated in Figure 9a. A slicer setting with a filling density of 100% is assumed. In this case, the distance between the tracks *T* is equal to the track width *W* and the track width *W* of the ideal droplet can be determined using Equation (Equation 1). The control of the freeformer is configured in such a way that the following droplet is deposited exactly at the distance of the track width *W*. This results in a square base area for a ideal droplet as shown in Figure 9a. With the help of the layer thickness LT, the ideal volume can be calculated according to
(4)$$\begin{array}{c} V_{\mathrm{Slicer},\mathrm{ideal}}=LT^{3}\cdot DAR^{2}. \end{array}$$

The frequency at which the droplets are deposited one after the other within a track depends on the printing speed *c* at which the build plate moves and the distance that must be covered before a new droplet is deposited. This distance is the track width *W*, so the slicer software calculates the frequency according to f=c/W. If the maximum frequency is exceeded, the slicer software reduces the printing speed. Consequently, the ideal volume flow is calculated according to
(5)$$\begin{array}{c} {\dot{V}}_{\mathrm{Slicer},\mathrm{ideal}}=V_{\mathrm{Slicer},\mathrm{ideal}}\cdot f=LT^{2}\cdot DAR\cdot c. \end{array}$$

As already noted in Section 4.1, a certain void volume is always present in the APF process due to the round shape of the droplets in the droplet chain. This void volume is shown in Figure 9b and must be taken into account for the real volume flow “requested” by the slicer software. The void volume is taken into account by the residual porosity factor *K*, which is defined as follows:(6)$$\begin{array}{c} K=\frac{V_{\mathrm{Slicer},\mathrm{void}}}{V_{\mathrm{Slicer},\mathrm{ideal}}}. \end{array}$$

Two extreme cases can be considered. At K=0 there is no more air inside, so the ideal volume is completely filled and the density of the ideal volume corresponds to the bulk density of the material used. On the contrary, at K=1 the ideal volume is exclusively filled with air and its density thus corresponds to that of air. The real volume of a droplet is the difference between the ideal volume and the void volume and thus the real volume flow requested by the slicer software is calculated according to
(7)$$\begin{array}{c} {\dot{V}}_{\mathrm{Slicer},\mathrm{real}}=V_{\mathrm{Slicer},\mathrm{real}}\cdot f=LT^{2}\cdot DAR\cdot c\cdot \left(1-K\right). \end{array}$$

On the hardware side, there is also a volume flow. This volume flow of the freeformer occurring in the manufacturing process results from the frequency of the closable nozzle and the droplet volume:(8)$$\begin{array}{c} {\dot{V}}_{\mathrm{freeformer}}=f\cdot V_{\mathrm{Droplet}}. \end{array}$$

The droplet volume results from the discharged plastic volume, which is considered as an inflexible volume. This volume can be calculated via the diameter of the plasticating cylinder $D_{\mathrm{Plast}}$ and a certain travel increment *L* of the screw that conveys the volume through the nozzle. This travel increment *L* is given by the DR and a reference travel increment $L_{0}$ defined by $DR=L/L_{0}$. According to Arburg, this reference travel increment $L_{0}$ is 0.00115 μm %^−1^. Overall, the volume flow of the freeformer is calculated according to:(9)$$\begin{array}{c} {\dot{V}}_{\mathrm{freeformer}}=f\cdot DR\cdot L_{0}\cdot \frac{\pi }{4}\cdot D_{\mathrm{Plast}}^{2}. \end{array}$$

On the hardware side, the frequency is regulated according to f=c/W. If the two volume flows from Equations (Equation 7) and (Equation 9) are set equal, the frequency can be shortened and the relation
(10)$$\begin{array}{c} LT^{3}\;\cdot \;DAR^{2}\;\cdot \;\left(1-K\right)=DR\;\cdot \;L_{0}\;\cdot \;\frac{\pi }{4}\;\cdot \;D_{\mathrm{Plast}}^{2} \end{array}$$
is obtained between the parameters LT, DAR, and DR. The free variables of Equation (Equation 10) are LT, DAR, DR and the desired porosity of the part, which is defined in K. One of the three parameters LT, DAR, or DR can therefore be calculated for a desired porosity. For example, for any LT-DAR combination, the DR for optDF can be calculated if the residual porosity at the state of optDF is known.

### 4.4. Validation of the Model

The model derived in the last chapter can be validated by comparing the calculated residual porosities with those determined experimentally. For the five parameter sets where optDF was reached, the density was measured by using the principle of Archimedes and the porosity using a µCT. These values can be found in Table 4. Since the residual porosity factor *K* refers to an representative volume element (RVE) from Figure 9 while the measured values are determined global, the assumption is made that the entire volume of the part is uniformly composed of individual RVEs. The measured density can be converted into a porosity by using the following equation
(11)$$\begin{array}{c} K=-\frac{\rho_{\mathrm{Specimen}}-\rho_{0}}{\rho_{0}-\rho_{\mathrm{Air}}}, \end{array}$$
where $\rho_{\mathrm{Specimen}}$ represents the density of the specimen, $\rho_{0}$ represents the bulk density of ABS, and $\rho_{\mathrm{Air}}$ represents the density of air in the voids. Since the density of air is much smaller than of ABS, it can be neglected. Consequently, a residual porosity factor *K* is given for the density measurement according to the principle of Archimedes and µCT.

By using Equation (Equation 10), the residual porosity factor *K* can be calculated by reforming the equation and use LT, DAR and DR according to the five parameter sets at optDF from Table 4. Figure 10 compares the three residual porosity factors determined theoretically and experimentally. The difference between the two experimental methods can be explained as follows: For density measurement, small air bubbles that adhere to the rough surface of the specimen can reduce the density and thus increase the porosity. However, this influence is considered to be small. In the case of porosity determination with µCT, the threshold between material and air influences the size of the porosity. This influence was in the order of ±1%. Despite these method-related errors, the measurements are in good agreement with each other.

If the two experimental residual porosity factors are compared with the theoretically determined one, a certain deviation is noticeable. This deviation could be explained by the fact that process-related variations of LT, DAR, and DR are not considered in the model. In the manufacturing process, a certain deviation between real and ideal LT and DAR can be assumed due to the positioning accuracy of the axes, as well as a variation of DR. Hirsch et al. have shown that the parameter DR varies during the manufacturing process [3]. Figure 10 shows how an absolute DR deviation of ±1% during the manufacturing process in relation to the DR setting affects the residual porosity factor *K*. Taking this process related deviation and the method-related errors into account, the theoretically determined residual porosity factors are in good agreement with the experimentally determined ones. The model can therefore be used to calculate suitable parameter sets for a desired degree of filling. Since the influence of nozzle temperature was not investigated, it is not considered in the model. Further investigations would be useful.

Another aspect can be derived out of Figure 10. The magnitude of the deviation of the residual porosity factor *K* decreases with increasing DR. This effect can be attributed to the change in volume flow, which varies in intensity depending on the DR. With Equation (Equation 9), the ratio of two volume flows is given by ${\dot{V}}_{1}/{\dot{V}}_{0}=DR_{1}/DR_{0}$. For the LT-DAR combination LT150DAR14, a DR that increases from 32% to 33% within a layer would fill this layer by 3.13% more. In contrast, for the LT-DAR combination LT250DAR14, a DR variation from 147 to 148% would only cause an increase in volume flow of 0.68%. Therefore, for small DR, process-related deviations in DR have a higher impact on the degree of filling than for a large DR.

### 4.5. Optimal Degree of Filling and Different Residual Porosity

In the two chapters above, a first volume flow modelling of the APF process was proposed and for the validation the different measured residual porosity was taken into account. This difference is only noticeable by the requirement for a maximum part density while maintaining geometric accuracy, since, as discussed in Section 4.1, equal part densities can be achieved for each LT-DAR combination. Based on the 3D reconstructions from the µCT, it was possible to determine not only how large the residual porosity is at optDF but also what the shape and distribution of the porosity is. In Figure 6, the appearance of the voids for the five parameter sets is visible. It is noticeable that almost no voids occur within a layer for each of the parameter sets. This observation is in good agreement with the expectation of optDF. Upon reaching this state, there are no gaps between the tracks not only on the top of the surface, but also in the inner layers. Only in the transition from perimeter to filling small voids can be found. However, the reason for the different residual porosity depending on the parameter sets can be found between the interfaces of two layers. Depending on the parameters LT and DAR, the shape of the droplet inside a part and the associated void volume seem to be different.

If the model is used for a LT-DAR combination to determine the DR at optDF, the residual porosity factor *K* is needed. Based on the measured densities from Table 4 at optDF, the dependence of the residual porosity factor *K* on LT and DAR is given in Figure 11.

Related to the earlier predictions of Charlon et al. or Hentschel et al. that an increase in density occurs as LT or DAR decreases, fails at this point [8,10]. In Table 5, three explanations for the difference in residual porosity are considered.

1. An intuitive explanation might be that the total amount of possible void locations correlates with the residual porosity. This was estimated under “Contact surface” by calculating the total contact surface between the tracks within the layers and between the layers. This explanation does not fit the behaviour of the residual porosities from Table 4. In Figure 6, it can be seen that the size of the voids varies depending on the parameter set, which is probably the reason why the explanation “Contact surface” does not work.

2. Another possible explanation was taken by McDonagh et al. [12]. They notice that the merging of the tracks is enhanced for a 10% larger DAR compared to the measured DAR of a droplet chain. After previous investigations, the DAR of the droplet chain can be determined by using Equation (Equation 1) and empirical equations: For the machine parameters from Table 3, the length of the droplet is given by $D_{\mathrm{Length}}=\left(2\cdot DR+135\right)$ μm and the width of the droplet is given by $D_{\mathrm{Width}}=\left(0.16\cdot DR+235\right)$ μm. As seen in Table 5 under “Compensate merging”, the ratio from DAR to DAR of the droplet chain does not correspond to 1.1, so this explanation cannot explain the difference in residual porosity either.

3. In this work, a new explanation is proposed. For example, for the investigated parameter set LT150DAR14DR32, the droplet length according to the above empirical equation is 199 μm and the desired LT value is only 150 μm. Assuming the droplet has a similar shape to the available RVE from Figure 9, it will approximately form as a sphere. Otherwise, a force acts on the droplet and deforms it. As a result, the same volume spreads in a more adapted way and fills the available space under less void formation. The more a droplet has to deform, the more voids can be compacted. If a ratio is calculated from the droplet length and the LT, and from the droplet width and the track width, the average of these two ratios can be used as a criterion for the deformation of a droplet. The correlation between the “Deformation factor” and the residual porosity from Table 4 is in good agreement. Further investigations are useful in the future for validating this explanation.

Finally, the existing approach for parameter determination is compared with the new approach presented in this work. With regard to the parameters DR, DAR and LT, the existing approach is as follows [6,12]: With a suitable nozzle temperature, droplet chains at a desired DR are generated and the droplet length as well as the DAR of a droplet chain must be measured. The LT should be approximately correspond to the droplet length. Starting from the measured DAR of a droplet chain for the DAR in the slicing software, several specimens with varying DAR are manufactured. This approach is illustrated in Figure 12b for an constant LT and DR. If the DAR is decreased, the density of a specimen increases until a desired degree of filling is reached. Although a part with an optDF can be achieved, this does not necessarily have to be an optDF with low residual porosity. Using this approach, for example, Hirsch et al. conclude that the lower the DAR value, the higher the density and the mechanical properties [3].

In comparison, Figure 12a shows the approach used in this work. The DR is varied for constant DAR and LT. In this case, an optDF can be determined for one LT-DAR combination and searched for further optDF on other combinations. However, a more appropriate approach for parameter identification is the determination of the residual porosity factor *K* as a function of LT and DAR and the use of the proposed model.

### 4.6. Mechanical Properties at Optimal Degree of Filling

It was shown in Section 3.5 that tensile strength and tensile modulus correlate with density of the dog-bone specimens, as already described by Charlon et al. [8]. Different densities were measured for the five parameter sets at optDF from Figure 7d. These differences correspond to those obtained for the cuboid specimens manufactured with the same parameter sets. The density of the dog bone specimens, however, are on average about 0.02 g/cm^3^ lower than the values of the cuboid specimens. An explanation for this could be the different surface to volume ratio of the two specimens types. The dog-bone specimens have a larger surface area in relation to their volume than the cuboid specimens. For this reason, the voids created between the perimeter and the filling have a larger impact on the residual porosity of the dog bone specimens. Regardless of the shape of the part, it is observed that the densities and related mechanical properties in the state of optDF differ significantly for the different parameter sets. For optimization of mechanical properties, a careful consideration of optDF must be taken into account. Finally, the larger variation in density and mechanical properties for LT150DAR14DR32 compared to the other parameter sets can be explained by the process-related variations of the DR discussed in Section 4.4. A parameter set with small DR is therefore not recommended.

## 5. Conclusions

Since the Arburg Plastic Freeforming is a relatively new additive manufacturing process, knowledge of the influences on volume-filling of a part, and in particular the mechanisms, is limited. This work demonstrates how an equal degree of filling or density can be achieved for different parameter sets, consisting of the layer thickness, the droplet aspect ratio, and the discharge rate. These results contradict the present state of the art, which implies that high density can only be achieved at low layer thickness and droplet aspect ratio. If the condition of geometric accuracy is taken into account in addition to the objective of a high degree of filling, exactly one state of optimal degree of filling can be found for each combination of layer thickness and droplet aspect ratio. It turns out that the state can be determined from the minimum of the surface roughness.

Furthermore, the state of optimal degree of filling was investigated for different parameter combinations regarding the residual porosity and the mechanical properties. The residual porosity at optimal degree of filling differs depending on the layer thickness and droplet aspect ratio. A reason for the differences might be that the deformation of the droplets after leaving the nozzle is different and this affects how much the voids can be decreased in size. Additional research to understand this mechanism is desirable in the future. Since the residual porosity is different, also the mechanical properties differ. Consequently, the question raised in this work can be answered in such a way that different parameter sets lead to different optima between mechanical properties and geometric accuracy. These differences cannot be clarified with the usual explanation of previous works, whereby a low layer thickness and a low droplet aspect ratio reduce the residual porosity to a minimum.

In order to find the optimal degree of filling for any parameter combination, a model was proposed and validated. Based on this, the following key points are highlighted:A high absolute discharge rate increases the process stability against variations of the discharge rate.A high droplet aspect ratio increases the deformation of the droplets and thus reduces the residual porosity at optimal degree of filling.A high layer thickness reduces the printing time and probably has a small influence on the deformation factor. However, a higher surface roughness must be taken into account.

To summarise, an example is given. For a part with the highest possible mechanical properties, high surface quality while maintaining process stability, the following applies: discharge rate high, droplet aspect ratio high, and layer thickness as low as possible.

## Figures and Tables

**Figure 1 polymers-15-01516-f001:**
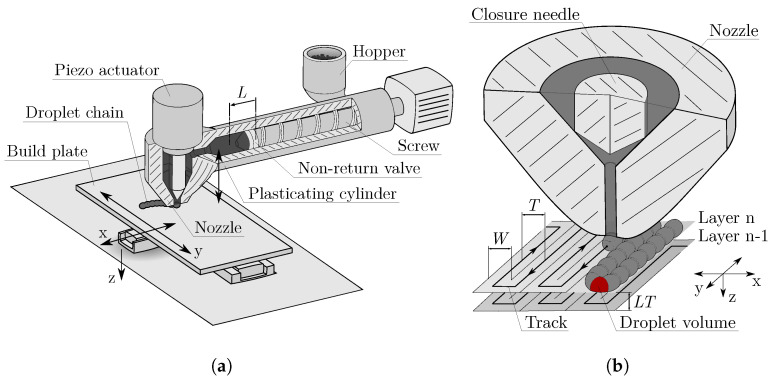
Visualisation of the APF process. (**a**): The most important components of the Arburg freeformer are given in a sketch of the moveable build plate with the three axes, the plasticating cylinder, and the closable nozzle. (**b**): Visualisation of important parameters that have an influence on the volume-filling of a layer. For simplicity, the orientation of the tracks is 0° orientation.

**Figure 2 polymers-15-01516-f002:**
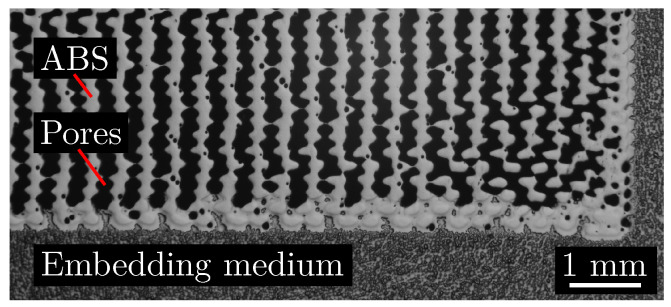
Microscopic image of a cross-section through an underfilled specimen embedded for the purpose of surface sealing. No embedding medium is entering the microstructure.

**Figure 3 polymers-15-01516-f003:**
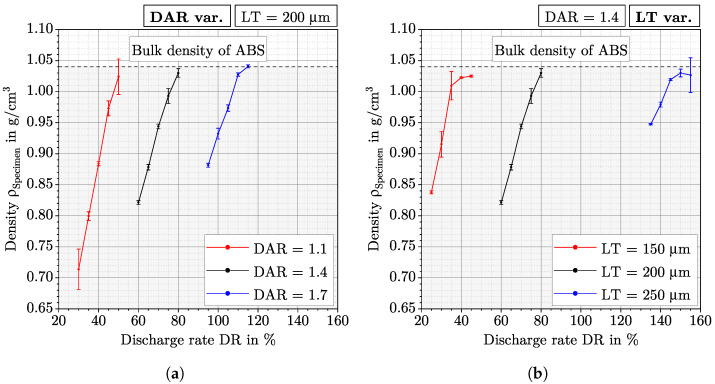
Measured density of all parameter sets from Table 1. For each LT-DAR combination the DR is changed in a range of 20%. The mean value and one standard deviation of the density are given for each parameter set. The bulk density for ABS is indicated by a horizontal line. (**a**): Density measurements for different DAR at constant LT. (**b**): Density measurements for different LT at constant DAR.

**Figure 4 polymers-15-01516-f004:**
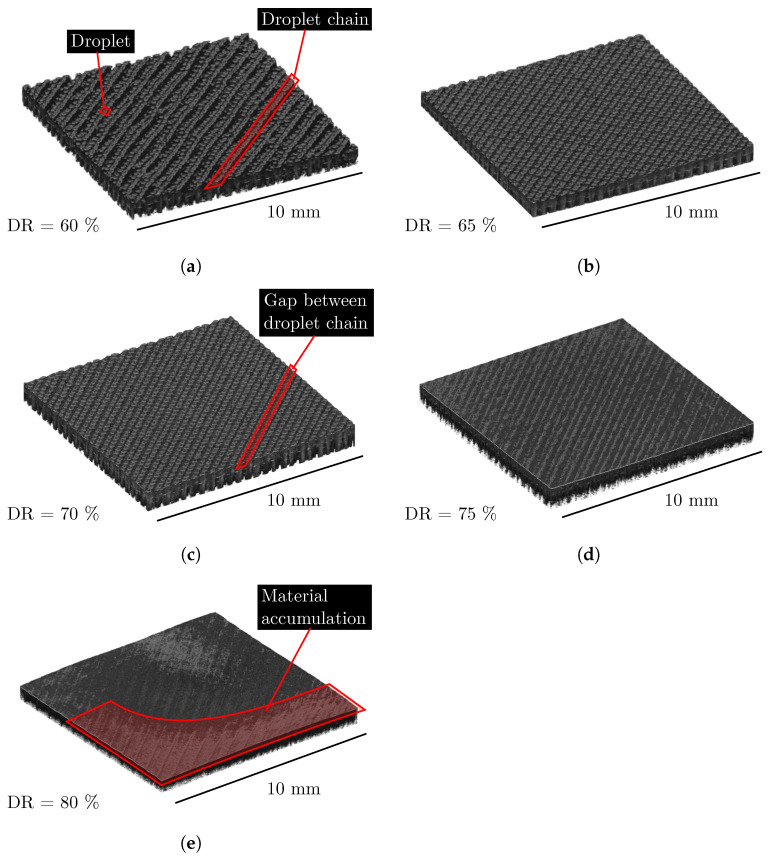
The 3D reconstructions of the surface structure from the top surface of the LT-DAR combination LT200DAR14. DR is varied between the individual 3D reconstructions. Depending on the degree of filling, the individual droplets and droplet chains are visible. (**a**–**e**): The five different DR settings.

**Figure 5 polymers-15-01516-f005:**
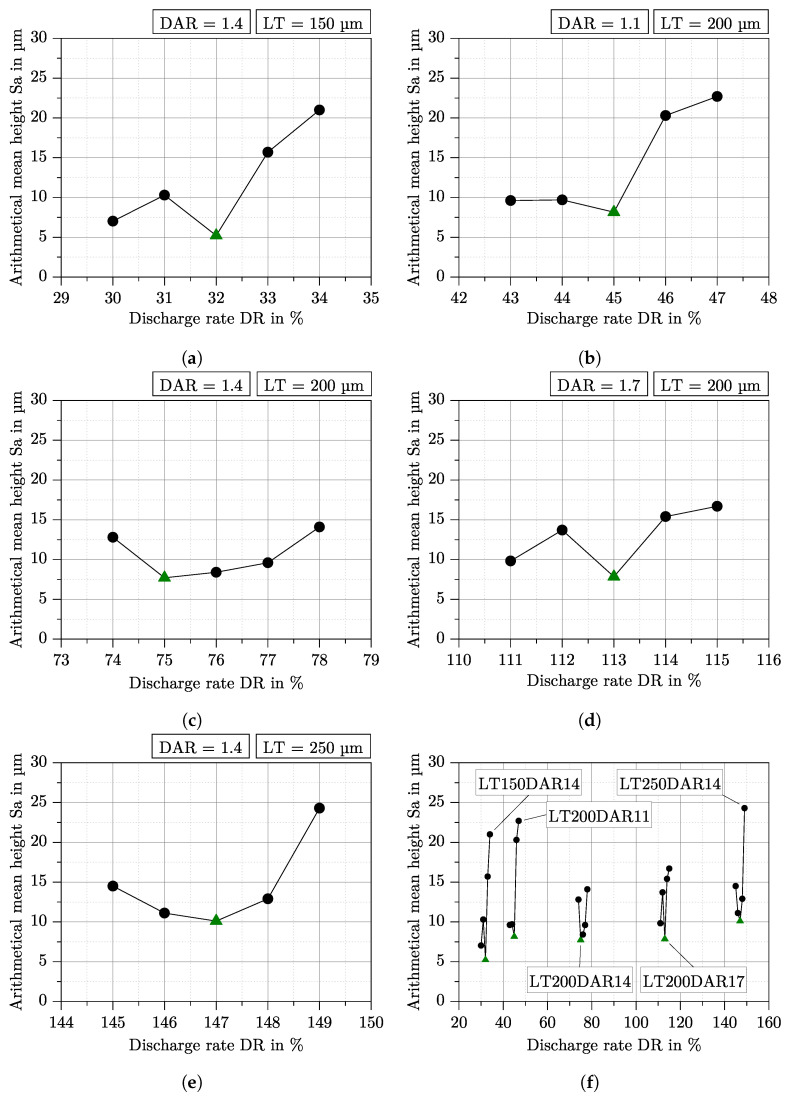
Arithmetical mean height Sa above the DR for the five LT-DAR combinations from Table 2. The range of DR is only 5% compared to the density measurement in Figure 3. The minimum of Sa is marked with a green triangle. (**a**–**e**): Sa for the individual LT-DAR combinations. (**f**) Comparison of all LT-DAR combinations.

**Figure 6 polymers-15-01516-f006:**
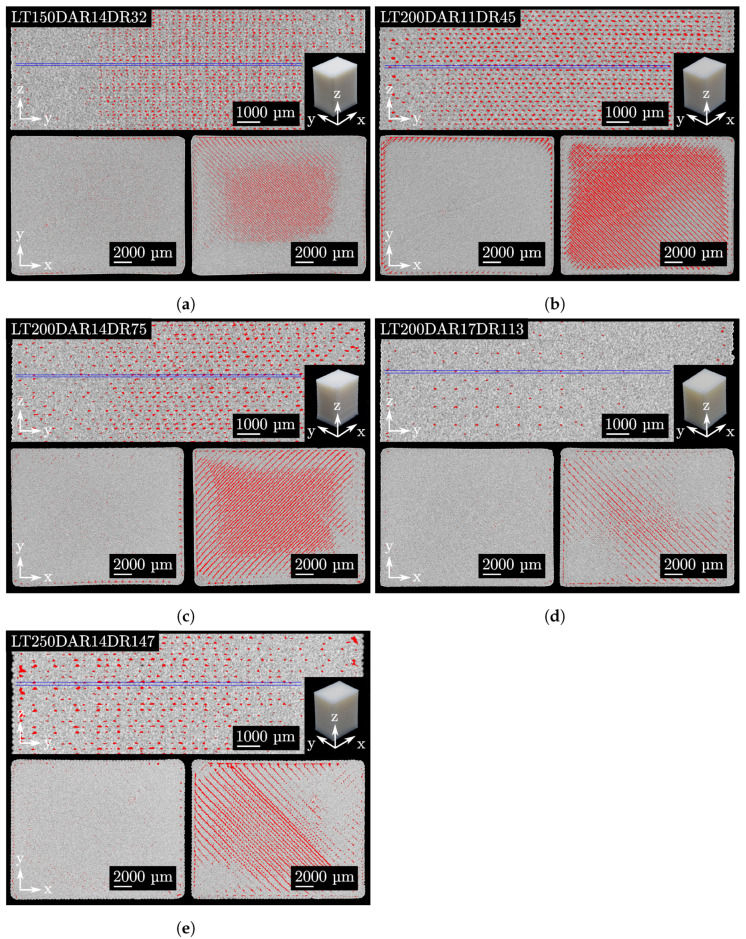
Slice images of the specimens with the specific parameter sets from Table 2. The left bottom corner shows a slicing plane within a layer, while the right corner shows a slicing plane between two layers. The top shows a slicing plane through multiple layers. The two slicing planes at the bottom are marked with two blue lines in the upper slicing plane. Porosity is coloured red. (**a**–**e**): The five different parameter sets.

**Figure 7 polymers-15-01516-f007:**
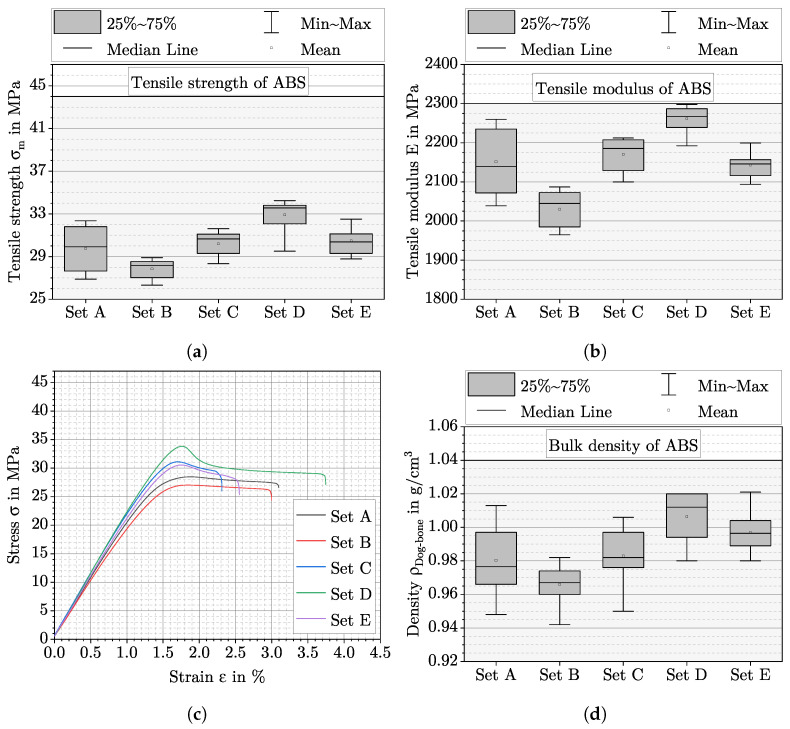
Mechanical properties and density of the dog-bone specimens with the specific parameter sets from Table 2. Set A = LT150DAR14DR32, Set B = LT200DAR11DR45, Set C = LT200DAR14DR75, Set D = LT200DAR17DR113, and Set E = LT250DAR14DR147. (**a**): Tensile strength, (**b**): tensile modulus, (**c**): one representative stress–strain curve for each parameter set, and (**d**): density for the specific parameter sets.

**Figure 8 polymers-15-01516-f008:**
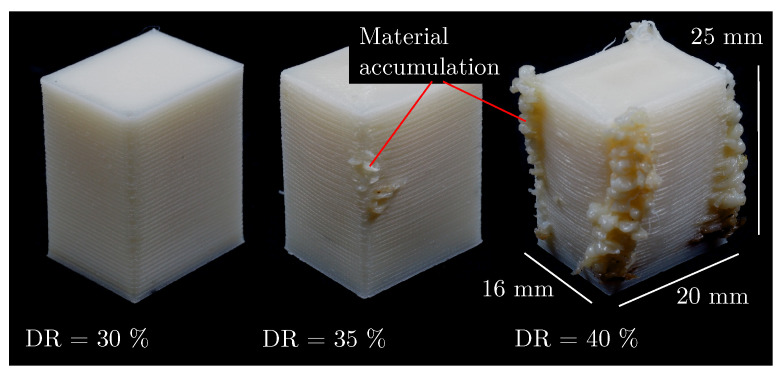
Three different specimens from the LT-DAR combination LT150DAR14. The DR is increased from the left to the right specimen and starting from a DR of 35% material accumulations are visible. The appearance of material accumulations is considered as the beginning of overfilling.

**Figure 9 polymers-15-01516-f009:**
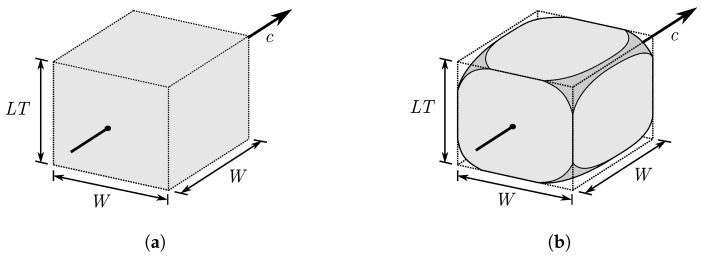
A representative volume element (RVE) is given for one droplet volume inside an additive manufactured part. (**a**): The ideal droplet volume in the slicing process. (**b**): The real droplet volume due to voids between the droplets.

**Figure 10 polymers-15-01516-f010:**
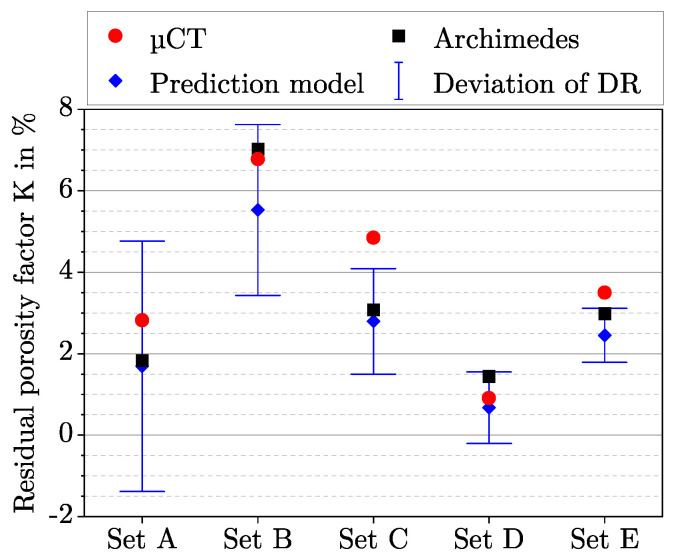
Comparison of the three residual porosity factors determined theoretically and experimentally for the parameter sets from Table 2. Set A = LT150DAR14DR32, Set B = LT200DAR11DR45, Set C = LT200DAR14DR75, Set D = LT200DAR17DR113, and Set E = LT250DAR14DR147. The abbreviation µCT and Archimedes refers to the two experimental methods. The error bar includes the model prediction in case of a statistical deviation of the DR by an absolute of 1%.

**Figure 11 polymers-15-01516-f011:**
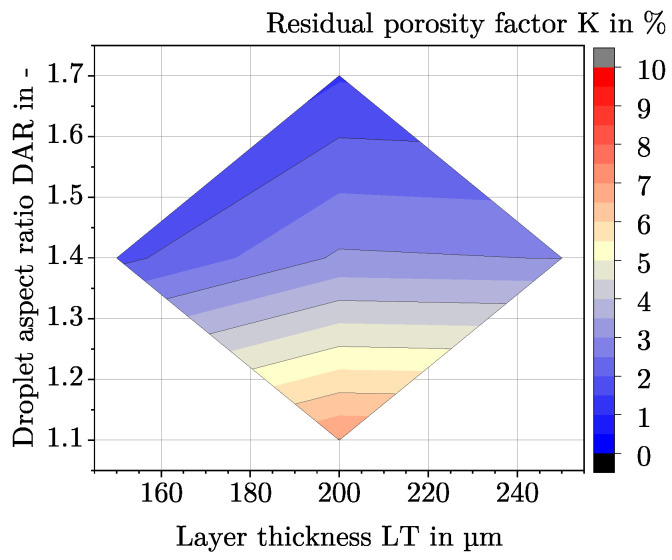
Dependence between the residual porosity factor *K* at optDF and LT and DAR. Between the five measured points from Table 4, the residual porosity factor is approximated.

**Figure 12 polymers-15-01516-f012:**
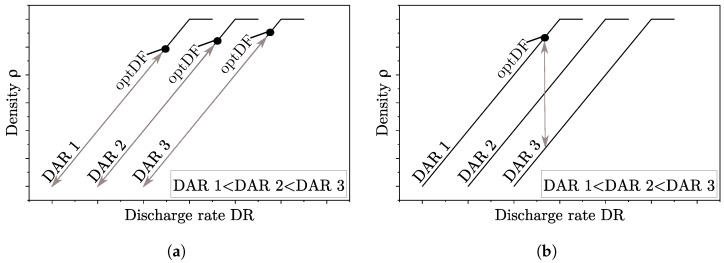
Comparison of the existing approach for parameter determination for optDF with the approach used in this work. The objective is presented in a qualitative diagram with constant LT that relates the density of a part and the DR. (**a**): Approach used in this work. (**b**): Existing approach.

**Table 1 polymers-15-01516-t001:** Overview of the five LT-DAR combinations and the parameter sets resulting with the selected DR. Specimens were manufactured for measurement of density and surface analysis.

LT-DAR Combination	Parameter Set	LT in μm	DAR in -	DR in %
LT150DAR14	LT150DAR14DR25	150	1.4	25
	LT150DAR14DR30			30
	LT150DAR14DR35			35
	LT150DAR14DR40			40
	LT150DAR14DR45			45
LT200DAR11	LT200DAR11DR30	200	1.1	30
	LT200DAR11DR35			35
	LT200DAR11DR40			40
	LT200DAR11DR45			45
	LT200DAR11DR50			50
LT200DAR14	LT200DAR14DR60	200	1.4	60
	LT200DAR14DR65			65
	LT200DAR14DR70			70
	LT200DAR14DR75			75
	LT200DAR14DR80			80
LT200DAR17	LT200DAR17DR95	200	1.7	95
	LT200DAR17DR100			100
	LT200DAR17DR105			105
	LT200DAR17DR110			110
	LT200DAR17DR115			115
LT250DAR14	LT250DAR14DR135	250	1.4	135
	LT250DAR14DR140			140
	LT250DAR14DR145			145
	LT250DAR14DR150			150
	LT250DAR14DR155			155

**Table 2 polymers-15-01516-t002:** For the determination and investigation of the optDF, specimens were manufactured according to the parameter sets in the table. Overview of the range of DR and the selected DR for the the five LT-DAR combinations from Table 1.

LT-DAR Combination	Determination of optDF	Investigation of optDF
	Range of DR in %	Selected DR in %
LT150DAR14	30–34	32
LT200DAR11	43–47	45
LT200DAR14	74–78	75
LT200DAR17	111–115	113
LT250DAR14	145–149	147

**Table 3 polymers-15-01516-t003:** Overview of the machine parameters used for the freeformer. All parameters were kept constant for the manufacturing of the specimens.

Parameter	Value	Unit
Temperature nozzle	260	°C
Temperature zone 2	200	°C
Temperature zone 1	170	°C
Chamber temperature	100	°C
Melt cushion	1.5	mm
Dosing stroke	4	mm
Decompression speed	2	mm/s
Decompression distance	5	mm
Circumferential speed	4	m/min
Stagnation pressure	50	bar

**Table 4 polymers-15-01516-t004:** Measured density and porosity for the specific parameter sets from Table 2. With the help of the bulk density of ABS, the measured density is converted into a porosity.

Specific Parameter Set	Measured with Principle of Archimedes		Measured with μCT
	Density in g/cm^3^	Porosity in %	Porosity in %
LT150DAR14DR32	1.021	1.83	2.82
LT200DAR11DR45	0.967	7.03	6.78
LT200DAR14DR75	1.008	3.08	4.85
LT200DAR17DR113	1.025	1.44	0.91
LT250DAR14DR147	1.009	2.98	3.50

**Table 5 polymers-15-01516-t005:** For the difference in residual porosity between the parameter sets, three explanations are presented with abbreviations for the concept.

Parameter Set at optDF	Contact Surface	Compensate Merging	Deformation Factor
	in mm^2^	in -	in -
LT150DAR14DR32	≈90,800	1.16	1.24
LT200DAR11DR45	≈75,300	1.02	1.11
LT200DAR14DR75	≈67,600	1.62	1.15
LT200DAR17DR113	≈62,600	2.42	1.27
LT250DAR14DR147	≈53,900	2.32	1.23

## Data Availability

The data presented in this study are available on request from the corresponding author.

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
