# Peer review of "Can Different Parameter Sets Lead to Equivalent Optima between Geometric Accuracy and Mechanical Properties in Arburg Plastic Freeforming?â€"

_polymers, 2023, doi:10.3390/polym15061516_

Round 1
Reviewer 1 Report
This manuscript should be completely rewritten because it is not in the form of a thesis.
1)
In Abstract: Two-thirds are written about background and previous works, and the description for this research is insufficient. For instance, there is no description of concrete experimental results and concept of optimal design. Since no explanation is seen for the Arburg Plastic Freeforming process, it is impossible to understand the porosity and droplets mentioned in the Abstract.
2)
The discharge rate is important factor for optical design. However, this manuscript does not provide a detailed description of the discharge rate required to facilitate understanding of the series of experimental results.
3)
The object of this study should be written at the end of the Introduction.
4)
It is very difficult to read this manuscript because the Results and Discussion are written in different places. I think it is better to merge the Discussion in Section 4 with the Experimental results for Figures 3-7 to prevent the confusion.
5)
The author should mention in 3.2 what can be inferred from the linear behavior shown in Figure 3 and the significance of this result for the optimal design.
6)
The authors should mention in 3.2 about why the surface structure changes with the discharge rate.
7)
The authors should mention in 3.4 and 3.5 about what is optimal degree of filling. Due to the lack of the explanation in 3.4 and 3.5, it is difficult to understand the significance of this study.
8)
The stress-strain behavior is usually drastically changed by existence of voids. However, the difference may be difficult to understand by the values of tensile strength and tensile modulus shown in Figure 7. It is better to discuss the difference of the mechanical properties by adding representative stress-strain curves used to obtain Figure 7.
Reviewer 2 Report
In this study, the joint influence of three parameters (LT, DAR and DR) on printing quality of APF, which broke through the previous researches on the influence of a single parameter. I have two questions.
1. The droplet aspect ratio (DAR) represents the morphology characteristic of droplet, which is not a direct control parameter. How does the user set DAR?
2. In the case of overfilling as shown in Fig.4e, why is material accumulation uneven?
Reviewer 3 Report
This article presents an experimental / analytical approach in determining various process parameters that provide geometric accuracy and better mechanical properties in Arburg Plastic Freeforming process. The process parameters of layer thickness, droplet aspect ratio and discharge rate are used as independent variables to investigate various results such as density, tensile strength, and modulus of elasticity. In Section 4, a further evaluation and discussion on optimal degree of filling were made. The methods are technically sound, and very well explained in the manuscript. The results are discussed satisfactorily. The study has merit. It is a well-structured and generally well-written article, slightly on the longish side. Authors are recommended to address the following comments.
- In table 5, a "comma" is used as decimal separator. Use of "point" would be preferable in order to be consistent throughout the text.
- Occasionally a few typos are detected. A proofreading is recommended.
Round 2
Reviewer 1 Report
The previous manuscript was revised completely and improved by taking into accout the Reviewers' comments. This manuscript has interesting concept, so it is worth for publication.